# The Role of Glial Cells in Different Phases of Migraine: Lessons from Preclinical Studies

**DOI:** 10.3390/ijms241612553

**Published:** 2023-08-08

**Authors:** Marta Vila-Pueyo, Otilia Gliga, Víctor José Gallardo, Patricia Pozo-Rosich

**Affiliations:** 1Headache and Neurological Pain Research Group, Vall d’Hebron Institute of Research (VHIR), Universitat Autònoma de Barcelona, 119-129 Passeig de la Vall d’Hebron, 08035 Barcelona, Spain; 2Headache Unit, Neurology Department, Vall d’Hebron University Hospital, 08035 Barcelona, Spain

**Keywords:** migraine, glia, astrocyte, microglia, satellite glial cell, cortical spreading depolarization, orofacial pain, headache, chronic migraine, migraine chronification

## Abstract

Migraine is a complex and debilitating neurological disease that affects 15% of the population worldwide. It is defined by the presence of recurrent severe attacks of disabling headache accompanied by other debilitating neurological symptoms. Important advancements have linked the trigeminovascular system and the neuropeptide calcitonin gene-related peptide to migraine pathophysiology, but the mechanisms underlying its pathogenesis and chronification remain unknown. Glial cells are essential for the correct development and functioning of the nervous system and, due to its implication in neurological diseases, have been hypothesised to have a role in migraine. Here we provide a narrative review of the role of glia in different phases of migraine through the analysis of preclinical studies. Current evidence shows that astrocytes and microglia are involved in the initiation and propagation of cortical spreading depolarization, the neurophysiological correlate of migraine aura. Furthermore, satellite glial cells within the trigeminal ganglia are implicated in the initiation and maintenance of orofacial pain, suggesting a role in the headache phase of migraine. Moreover, microglia in the trigeminocervical complex are involved in central sensitization, suggesting a role in chronic migraine. Taken altogether, glial cells have emerged as key players in migraine pathogenesis and chronification and future therapeutic strategies could be focused on targeting them to reduce the burden of migraine.

## 1. Introduction

### 1.1. Migraine

Migraine is a complex and debilitating neurological disease that affects 15% of the population worldwide [1]. It is defined by the presence of recurrent severe attacks of disabling headache accompanied by other neurological symptoms such as nausea, photophobia, and allodynia, which lasts 4–72 h and during which the sufferer is completely unable to function normally [1]. Migraine is the second-most disabling chronic neurological disease [2] that causes a poor quality of life with an associated social and economic burden due to healthcare and treatment costs, work absenteeism and reduced productivity [3].

Clinical and preclinical advances have substantially improved the understanding of the pathophysiology of migraine attacks, linking them to an activation and sensitization of the trigeminovascular system [4], as well as brainstem and diencephalic nuclei [5,6]. The trigeminovascular system is composed of the trigeminal ganglia (TG), which are formed by the soma of neurons that innervate the meninges and large cerebral arteries, and that send central axons to trigeminocervical complex (TCC) neurons located in the upper cervical cord [7]. TCC neurons convey nociceptive signals to neurons of the brainstem, hypothalamus and thalamus, which relay the information to cortical areas [8]. These nuclei also modulate the activity of the trigeminovascular system, having an impact on the susceptibility to migraine too [9].

Moreover, calcitonin gene-related peptide (CGRP) has been identified as a key neuropeptide for migraine pathophysiology due to its clinical relevance [10,11]. In fact, CGRP is released during migraine attacks [12] and its infusion can trigger migraine in patients [13]. Anti-CGRP treatments have proven to reduce the number of headache and migraine days per month, to reduce acute medication use and to reduce migraine-related burden [14,15]. Currently, both anti-CGRP monoclonal antibodies and oral gepants have been approved by FDA and EMA.

Besides the important advancements performed to date to understand the pathophysiology of migraine attacks, the pathogenesis of migraine remains poorly understood. Hence, although it is known that the activity of the trigeminovascular system is enhanced in migraine patients and that their levels of CGRP are higher compared to healthy controls, the mechanisms that cause this higher activation and expression are still unknown. Moreover, little is known on the underlying mechanisms of the chronification of the disease, in which patients suffer from more than 15 days/month of severe headache [16].

### 1.2. Glial Cells

Glial cells, also named glia or neuroglia, are the most abundant cells within the nervous system. Although initially they were considered as passive supporting cells for neurons, it is now well accepted that they play active roles in the development and function of the nervous system. Among other functions, they maintain neural homeostasis by nurturing and enhancing neuronal function and by keeping a proper chemical environment. They also protect and assist in the repair and regeneration of neurons during injury, inflammation and infection [17]. Moreover, glial cells are capable of releasing neurotransmitters and growth factors via the classic fusion of secretory vesicles that influence the activity of neurons or other cells, from the immune system for instance, and of communicating with each other via gap junctions and calcium waves.

There exist different subtypes of glial cells, each one with distinct functions (Figure 1). In the central nervous system (CNS), glia include two major subtypes: macroglia and microglia [18]. Macroglia derive from neural stem cells and include astrocytes, oligodendrocytes and NG2-glia. Microglia, instead, have a myeloid origin and derive from foetal macrophages. In the peripheral nervous system (PNS), glia include Schwann cells, satellite glial cells, olfactory ensheathing cells and enteric glia [19]. Although each type of glial cell participates in a myriad of functions, astrocytes are mainly involved in maintaining the homeostasis of the CNS. Oligodendrocytes and Schwann cells are in charge of myelinating and supporting axons, whereas NG2-glia are lifelong precursors of oligodendrocytes. Microglia are the immune cells of the CNS and satellite glial cells support neurons within peripheral ganglia. Finally, olfactory ensheathing glia are lifelong regenerators of olfactory axons and enteric glia are in charge of supporting neurons in the enteric nervous system of the gastrointestinal tract.

### 1.3. Overview of the Role of Glia in Neurological Diseases

Glia are responsible for maintaining an homeostatic environment within the nervous system and participate in several processes that are essential for its correct development and function. Moreover, glia suffer morphological, transcriptional and functional changes in disease, a process that has been extensively characterized in astrocytes (astrogliosis) and microglia (microgliosis), highlighting the existence of an important contribution of glia in neurological disorders. Actually, there is a growing body of evidence that shows the implication of the different types of glial cells in a wide range of pathological conditions, including neurodevelopmental, neurodegenerative and neuropsychiatric disorders and in different pain conditions. As an example, current evidence indicates an active role of astrocytes both in the initiation and progression of epilepsy, probably due to an enhanced hyperexcitability linked to astrocyte dysfunction [20,21]. Also, several studies show that glial cells participate in the pathogenesis and chronification of pain [22,23,24]. In this sense, microgliosis and astrogliosis occur after nerve injury in neuropathic pain models, with microglia being more involved in the initiation of pain and astrocytes in the transformation from acute to chronic pain and in the maintenance of chronic pain.

Some of the neurological disorders where glia are relevant for their pathophysiology have shared mechanisms with migraine. For instance, migraine has been hypothesised to be associated with brain hyperexcitability, a characteristic seen in patients with epilepsy that, as mentioned above, has been linked to astrocyte dysfunction [25,26,27]. Chronic migraine shares several important features with chronic pain too, and the latter has been shown to have a strong glial component [28]. Moreover, migraine comorbidities, such as depression and stress, have also been suggested to have a contribution of glial cells [29]. This has led to the hypothesis that glial cells may be key players both in the pathogenesis and chronification of migraine and has promoted the development of studies to analyse this association.

This narrative review aims to give an overview of the current knowledge, although scarce, on the role of glial cells in migraine pathophysiology, focusing on three different phases of migraine: the aura phase, the headache phase and the chronification of the disease, through the analysis of the preclinical studies performed to date. A summary of the main findings is shown in Table 1. Moreover, therapies focusing on targeting glia will be discussed to uncover potential future therapeutic targets that could be relevant for migraine.

## 2. Role of Glia in Cortical Spreading Depolarization: Implications for the Migraine Aura Phase

Approximately 30% of migraine sufferers experience transient neurological disturbances manifested as visual, sensory or motor symptoms, that usually occur before the headache: the migraine aura [54]. Cortical spreading depolarization (CSD), considered the neurobiological correlate of migraine aura [55], is a short-lasting neuronal and glial depolarization wave that moves across the brain cortex followed by a wave of depression of evoked and spontaneous electroencephalogram activity [56].

An important role for astrocytes in CSD has been suggested since the publication of early studies that implicated glial membrane depolarization as a primary driver of the depolarization wave [57]. Astrocytes can communicate through propagated increases in intracellular calcium concentration: the astrocyte calcium waves [58]. These have temporal and spatial characteristics that are similar to CSD [30] and preclinical studies have shown the existence of astrocyte calcium waves along with waves of neuronal activation during CSD, suggesting a direct link between cortical astrocytes and CSD [31]. However, there exists conflicting evidence in regard to the role that these cells may have during CSD, as the pharmacological inhibition of astrocyte calcium waves does not terminate the propagation of CSD, but instead inhibits the associated vascular changes associated with CSD [32], suggesting a role for astrocytes in the vascular response, but not in the propagation of CSD.

Other evidence, however, suggests a crucial role for astrocytes in CSD. Mutations in the Na^+^, K^+^-ATPase gene (ATP1A2), which is mainly expressed in astrocytes, causes familial hemiplegic migraine type-2 (FHM2), a rare subtype of migraine that is clinically characterised by prolonged aura episodes [59]. This astrocytic protein is involved in the clearance of extracellular potassium and glutamate, a phenomenon that is critical in the susceptibility to CSD induction [33]. Besides the Na^+^, K^+^-ATPase, other astrocytic proteins that regulate extracellular glutamate concentration have also been found to be associated with an increased susceptibility to CSD, including the glial glutamate transporter 1 (GLT-1) [34] and aquaporin-4 (AQP4) [35]. Hence, taken altogether, the current data seem to suggest a role for astrocytes in the initiation, but not the propagation, of CSD that is based on the maintenance of a homeostatic extracellular glutamate concentration.

On the other hand, studies have shown that both acute and chronic CSD induce the process of reactive astrocytosis without the presence of neuronal injury [36,37]. Reactive astrocytosis, also known as astrogliosis, is a process by which astrocytes undergo a shift in form and function, affecting their release of cytokines and gliotransmitters, and is determined by an increased expression of glial fibrillary acidic protein (GFAP) [60]. Reactive astrocytosis has been shown 2 days after the induction of multiple CSDs for 3 h, or 1 week after once-daily CSD induction, and it persists for up to a few weeks [36,37]. Moreover, both studies seem to indicate the existence of a temporal association of CSD suppression with reactive astrocytosis, indicating that astrocytes undergo important morphological and functional changes during CSD, but that they return to baseline once CSD has terminated. These data support an active role for astrocytes in CSD.

Besides astrocytes, other types of glial cells have been shown to have a relevant role in CSD as well. Importantly, CSD activates microglia in rat cerebral cortexes [38,39], probably via the K^+^ inward rectifier (Kir) 2.1 [61]. Also, CSD induces microglial migration and motility, a change that could influence the electrical activity of the surrounding tissue, leading to higher susceptibility to CSD [40]. On another hand, microglia have also been involved in the generation of CSD. A study found that the depletion of microglia in slice cultures inhibits the induction of spreading depolarization, whereas the replacement of microglia in depleted cultures restores the susceptibility to spreading depolarization, suggesting an essential role for these cells in CSD initiation [41]. Moreover, the study correlated two different states of microglia polarization: M1, during which microglia produce pro-inflammatory cytokines and reactive oxygen and nitrogen species; and M2a, during which microglia produce anti-inflammatory cytokines and neurotrophic factors, with CSD susceptibility. This analysis showed a direct correlation between the microglial M2a state with a decreased susceptibility to CSD. Although the current data on microglia in CSD are still limited, they point towards an important role of this type of glial cell.

The role of satellite glial cells within the TG has also been explored in a mouse model of CSD. In this study, mouse TG were transcriptionally profiled at a single-cell resolution. Interestingly, among the different cell types of the TG, satellite glial cells and fibroblasts were the only ones found to be transcriptionally activated 1.5 h after a single CSD was mechanically induced [42]. These are interesting results as they show that CSD not only impacts glial cells within the cortex, but also distant regions of the nervous system. Further studies will have to be performed to understand the implication of these results and the role that satellite glial cells may have in this process.

To summarize, there exists important preclinical evidence showing a primary role of glial cells, especially astrocytes and microglia, in CSD initiation and/or propagation, with different molecular pathways implicated. Further studies are needed to better understand the role of these cells, and other types of glia, and whether they could be therapeutically targeted to inhibit the initiation and propagation of CSD.

## 3. Role of Glia in Orofacial Pain: Implications for the Migraine Headache Phase

The headache phase of migraine attacks is the most prominent, mostly due to the presence of pain and to the level of severity of the symptoms presented [62]. The study of the headache phase of migraine, through the analysis of orofacial pain mechanisms, has determined that the neuronal contribution is predominantly based on the peripheral sensitization of primary afferent neurons and the central sensitization in spinal cord and brain neurons [63]. Although the contribution of glial cells in the orofacial region has not been as well studied compared to the contribution of neurons and compared to the development of pain in other body regions, current data show an implication of the different types of glial cells in orofacial pain that may be of relevance to migraine.

In the orofacial region, neuropathic pain and chronic inflammatory pain models have been used in rodents to study the implication of neurons and satellite glial cells of the trigeminal ganglion in the development of orofacial nociception (for a review, see [64,65]). Neuropathic pain models affecting the infraorbital nerve or the inferior alveolar nerve induce changes in the activity of satellite glial cells, along with changes in their intracellular processes, such as the modification of the expression of p38 MAPK, ERK and phosphatases [66]. Also, trigeminal nerve injury induces ionic conductance changes in satellite glial cells that can be associated with ectopic discharges in trigeminal neurons [67]. Moreover, changes in the expression of satellite glial cell-specific proteins, such as the inward rectifying K^+^ channel Kir4.1, are crucial for the development of orofacial nociception both in the presence of neuropathic pain [68] or in naïve animals [69]. These studies highlight the role of satellite glial cells in different models of orofacial pain.

Interestingly, TG neurons can modify the activity of satellite glial cells through a variety of extracellular channels, including several purinergic channels, and via gap junctions as well [70]. Of relevance for migraine pathophysiology, TG satellite glial cells of rat express the canonical CGRP receptor (CLR/RAMP1), along with other CGRP receptors including the AM2 (CLR/RAMP3) and AMY3 (CTR/RAMP3) receptors [43,44]. Although satellite glial cells from naïve animals do not express CGRP, they do express its precursor (procalcitonin) and adrenomedullin as well [43], whose role in migraine pathophysiology has been hypothesised together with the other protein members of the CGRP family [71], besides not having any migraine effect when systemically administered to humans [72]. CGRP, however, has been found in satellite glial cells of an orofacial pain rat model, suggesting that its expression might be dependent on the development of nociceptive mechanisms [73].

The expression of the CGRP receptor in satellite glial cells supports the evidence that CGRP has an important role in modulating TG satellite glial cells’ function. In this sense, the neuronal release of CGRP upregulates the synthesis and release of nitric oxide (NO) from TG satellite glial cells via the activation of MAPK pathways [74], a mechanism that in turn modulates orofacial pain behaviours [73]. Interestingly, the glia-released NO stimulates neuronal CGRP expression through N-type calcium channels, building a CGRP-mediated positive feedback between TG neurons and satellite glial cells [73] (Figure 2), a process that could be relevant for the chronification of migraine and that could be blocked to have a therapeutic effect [46]. Actually, the therapeutic action of the recently approved antibodies against CGRP and its receptors could involve the disruption of abnormal neuron–satellite glial cell interactions, as these antibodies enter sensory ganglia but not the CNS [75]. On the same line, a different study showed that intra-TG administration of the glial inhibitor minocycline reverses CGRP-induced thermal nociception, reduces glial activity and downregulates the expression of relevant cytokines, supporting the role of satellite glial cells of the TG in orofacial nociception [76].

Besides CGRP, other migraine-relevant neuropeptides are released upon activation of trigeminal neurons [77]. One is the neurokinin substance P, which is expressed in small- to medium-sized TG neurons, and its receptor, NK1R, which is expressed in TG neurons, satellite glial cells and Schwann cells [45]. Interestingly, a link between substance P release in orofacial pain mechanisms and NK1-mediated activation of TG satellite glial cells has been suggested as data from in vitro studies indicate that substance P stimulates IL-1 production in glial cells [78,79] and that NK1R activation depolarizes the membrane potential in astrocytes [80]. Nevertheless, it is not likely that glial NK1R are essentially involved in migraine pain as targeting NK1R as an acute and/or preventative therapeutic approach for migraine failed in human trials, probably due to the fact that substance P and NK1R expression is lower in humans compared to rodents [81].

Pituitary adenylate cyclase-activating peptide 38 (PACAP-38) is also expressed both in neurons and satellite glial cells within the TG of humans and rats, whereas its receptors PAC1, VPAC1 and VPAC2 are expressed either in TG neurons or satellite glial cells, depending on the species [44]. Although PACAP-38 is an efficient inducer of migraine attacks in migraine patients, its role in orofacial pain in relation to glial cells has not been explored yet.

Taken altogether, the studies that have analysed the role of glial cells in orofacial pain in rodents, with a special focus on migraine-related mechanisms, point towards an important involvement of these cells, especially the satellite glial cells within the TG, in the initiation and maintenance of the headache phase in migraine. However, further studies are needed to better understand this role and to uncover potential new therapeutic targets focused on modulating satellite glial cell activity.

## 4. Role of Glia in Chronic Pain: Implications for Migraine Chronification

The chronification of migraine involves the transformation of the disease to its most severe form, in which patients suffer from more than 15 days/month of severe headache and the burden of the disease increases exponentially [16]. Although several risk factors have been associated with migraine chronification, including ineffective acute and preventive treatment, female sex, low educational status, stressful life events and the presence of comorbidities, the mechanisms behind this process are not understood [16,82]. The recent approval of different anti-CGRP treatments has proved beneficial for chronic migraine patients; nevertheless, there is still an important proportion of them who do not respond to any of the current treatments [83]. Thus, there is a need to better understand the process of migraine chronification to develop new therapeutic strategies that can benefit a wide majority of patients.

An important body of literature confirms the contribution of glial cells in the chronification of different pain conditions [23,24,84,85,86,87,88,89,90]. Moreover, reversing the injury-induced changes in glial cells using different approaches has been shown to reduce, or even abolish, pain behaviour in different rodent models [23,28,91]. This highlights the relevance of studying the role of glial cells in migraine chronification.

In the nitroglycerin (NTG)-induced chronic migraine model in mice [92,93], different markers of microglia activation have been found over-expressed in the TCC, including the purinoceptors P2X4R [47,48], P2Y12R [49], P2X7R [50], which are known regulators of central sensitization in inflammatory and neuropathic pain. These results were confirmed using specific antagonists of each receptor, 5-BDBD or MRS2395 and clopidogrel or Brilliant Blue G, respectively, which blocked the hyperalgesia induced by NTG. Other proteins that have been found over expressed in TCC microglia in chronic migraine mice include the glucagon-like peptide-1 receptor (GLP-1R) [94], which was previously found to inhibit neuropathic and cancer pain, and the sphingosine-1-phosphate receptor 1 (S1PR1) [95], whose blocking was known to relieve the development of chronic pain and to inhibit the activation of microglia.

Other studies have used the same chronic migraine model to analyse the inflammatory response of microglia in the process of central sensitization. In that sense, the NOD-like receptor protein 3 (NLRP3) inflammasome, which is an immune complex that regulates the maturation of IL-1β, and IL-1β itself were found over-expressed in microglia in the TCC of chronic migraine mice [51]. They also found that TCC neurons expressed the IL-1β receptor, suggesting that NLRP3 may mediate the inflammatory response in the central sensitization observed in chronic migraine and that there is an important neuron-glia cross-talk mediating this process. Another molecule, the microRNA miR-155-5p, was also studied to unravel the role of microglia in the chronic migraine model [52]. Interestingly, miR-155-5p was found overexpressed in the TCC of chronic migraine mice and its inhibition alleviated microglial activation and decreased the release of inflammatory substances.

On the same line, another group of studies has used specific drugs to modulate the activity of microglial cells in the chronic migraine model in mice. One such example is the administration of systemic minocycline to the chronic migraine mice, which decreased basal hind-paw allodynia, but did not alter the acute NTG-induced effect on allodynia [47]. In this study, minocycline also reduced the number of ionized calcium-binding adapter molecule 1 (Iba1)-labelled cells, which is a commonly used marker for microglia activation [96]. Another example is the administration of systemic roxadustat to the chronic migraine mice [53]. Roxadustat is a hypoxia-inducible factor-1α (HIF-1α) stabilizer that was found to reduce basal and acute NTG-induced hyperalgesia, to decrease inflammatory cytokine levels and to inhibit microglia activation, which was also measured through analysing Iba1 expression levels. However, an important limitation of using such drugs to modulate the activity of glial cells is that they are rather general glial inhibitors; so, when systemically administered, it is not possible to discern which glial cells are mediating the studied mechanisms. For instance, the studies using minocycline or roxadustat focused only on analysing changes in TCC microglia, but did not explore other regions that are also relevant for migraine pathophysiology, such as the TG, for example. Hence, they cannot rule out the possibility that the effects seen on allodynia might not be due to the effects of minocycline on the TCC, but on other regions of the CNS or PNS. Furthermore, the mechanisms of action of such drugs remain somewhat uncertain. Regarding minocycline, for instance, potential suggestions include the reduction of microglial production of proinflammatory factors [97], but further studies should be performed to better understand the mechanisms of action of these drugs.

Besides TCC microglia, TG satellite glial cells may also have a role in the chronification of migraine. This hypothesis is supported by the fact that satellite glial cells express the CGRP receptor and that there exists a CGRP-mediated positive feedback between TG neurons and satellite glial cells (for details, see the section “Role of glia in orofacial pain: implications for the migraine headache phase”) [73]. However, further studies should be performed to better understand this as the specific role of these glial cells in models of chronic migraine has not been analysed yet.

Taken altogether, these studies highlight the relevance that TCC microglia potentially have in the central sensitization of chronic migraine and that a better understanding of the function of these cells would be crucial to improve the current knowledge of chronic migraine and to develop new therapeutic targets.

## 5. Targeting Glia: Potential Future Therapeutic and Diagnostic Opportunities for Migraine

As evidenced by the studies analysed in the previous sections, glia are key factors in the development of some pathological processes related to migraine; their states could potentially modulate the susceptibility to cortical spreading depolarization, headache and migraine chronification, and, in turn, these migraine-related processes can also modify glial intrinsic characteristics. Glial cells are, therefore, a potential cell target for therapeutic approaches to treat neurological conditions, including migraine and its chronification.

Classic therapies targeting glia include the use of several different glial inhibitors [28]. Some of them include the above-mentioned minocycline, which among other non-specific actions reduces microglial production of proinflammatory factors including NO and IL-1β [98]; fluoroacetate and its metabolite fluorocitrate, that disrupt microglial and astrocytic metabolism via inhibition of the glial-specific aconitase [99]; propentophylline, that reduces the activity of microglia and astrocytes via cAMP upregulation [100]; L-α-aminoadipate, an astrocytic cytotoxin that reduces GFAP levels [101]; and ibudilast, a nonspecific phosphodiesterase inhibitor that decreases the glial release of pro-inflammatory cytokines and the development of gliosis [102]. Even though these inhibitors have been extensively used in the literature to improve the knowledge on glia, they pose several limitations that should be taken into account when analysing the outcomes of such studies. For instance, these compounds are not specific glial inhibitors. Instead, they affect other cellular processes as well that could be masking, or enhancing, the effects seen on glia. Hence, the negative outcomes of these drugs do not necessarily mean that glial cells are not involved in the mechanisms studied. Of note, ibudilast was tested in a double-blind, randomized, placebo-controlled trial in patients with chronic migraine without improving migraine symptoms or decreasing the frequency of attacks [103].

To overcome the limitations of using classic glial modulators, efforts are being made to develop glia-specific therapies to treat neurological disorders, which have been summarized in Table 2. One potential glia-specific therapeutic approach is the use of adeno-associated viral (AAV) vectors to deliver gene therapy specifically into the glial cells of interest [104]. Clinical trials have already shown the safety and effectiveness of using AAV vectors to specifically target cells in the nervous system, although to date, they have only been used to target neurons [104]. Preclinical studies, however, have successfully targeted and modulated glial cell activity using this approach, holding promise for future glial-specific therapies [105].

Another alternative approach to target glial cells is the use of nanoparticles [106]. These can be developed to cross the blood–brain barrier after systemic administration and to release their content into glia. Actually, preclinical studies have successfully used nanoparticles to specifically deliver small interfering RNAs (siRNAs) or mRNAs to either down- or upregulate the expression of a protein of interest in the targeted astrocytes. Although these studies seem promising for their potential in glial-specific drug delivery, further investigations are needed to refine and translate these tools as therapeutic options for neurological disorders including migraine.

Finally, cell-replacement therapy is garnering more attention in neurological diseases as an alternative therapy to modulate the activity of dysfunctional glial cells [107]. This therapy is based on preclinical evidence showing that transplanted healthy astrocytes can restore the appropriate homeostatic environment in the diseased nervous system, hence alleviating the symptoms of the disease. A promising example of this therapy was recently developed and tested for amyotrophic lateral sclerosis (ALS) patients [108]. In this clinical trial, human neural progenitor cells transduced with glial cell-derived neurotrophic factor (GDNF) differentiated to astrocytes were transplanted unilaterally into the lumbar spinal cord of ALS patients in a phase 1/2a study. There were no negative effects on motor function, and post-mortem tissue from treated patients showed graft survival and GDNF production. These results, along with other preclinical outcomes, show great promise to develop future strategies to target glial cells in neurological diseases, including migraine. However, one of the main limitations of using this approach in a condition like migraine would be the administration route. Intrathecal administration is a very invasive procedure; hence, other administration routes should be developed for cell-replacement therapy.

On another line, markers of glia are being used as biomarkers for the diagnosis of neurological disorders. For instance, blood levels of GFAP are used as a clinical test to evaluate mild traumatic brain injury, as they are correlated with clinical severity [109]. Improving the current knowledge on the roles of glial cells in migraine and unravelling the potential presence of glial biomarkers in migraine patients would be crucial, as currently there is a lack of biomarkers to diagnose migraine and to predict the response to anti-migraine treatments.

Taken together, evidence shows that, although classic glia modulators have been beneficial to improve the current knowledge and to modulate the activity of glial cells, more specific delivery systems, such as viral vectors and nanoparticles, and therapeutic approaches, such as cell-replacement therapy, are needed to overcome the limitations presented by classic compounds. Although preclinical, and scarce clinical, data are promising in regard to the effect that these new therapeutic strategies have on certain neurological conditions, further studies are required to be able to implement these strategies in migraine treatment.

## 6. Conclusions and Future Perspectives

There is supporting evidence from preclinical studies that glial cells may play a role in migraine pathophysiology, including in the aura phase, the headache phase and the chronification of the disease. The preclinical studies performed to date, which have been reviewed here, analyse the role of the different types of glial cells, including satellite glial cells, microglia and astrocytes, in different pathological processes that are of relevance for migraine. A summary of the potential role that glial cells may have in migraine can be found on Figure 3.

As reviewed here, the majority of the preclinical studies performed to date have used classical glial inhibitors to better understand the role of glia. However, as discussed above, these inhibitors have important limitations, such as a low specificity, which have a direct impact on the relevance of the outcomes of the studies. Currently, new strategies are being developed to overcome these limitations, including the use of viral vectors and nanoparticles to deliver the molecule of interest to specific glial cells and the use of cell-replacement therapy to replace dysfunctional glia. These new approaches will greatly improve the current knowledge on glia and, more specifically, on their role in migraine pathophysiology, which is currently deficient.

Future studies will determine whether glial cells are necessary to initiate migraine attacks and how relevant they are in increasing individual susceptibility to migraine. Moreover, we expect that an improved knowledge on the topic will attract growing attention to develop new migraine-specific therapies focused on targeting glia.

## Figures and Tables

**Figure 1 ijms-24-12553-f001:**
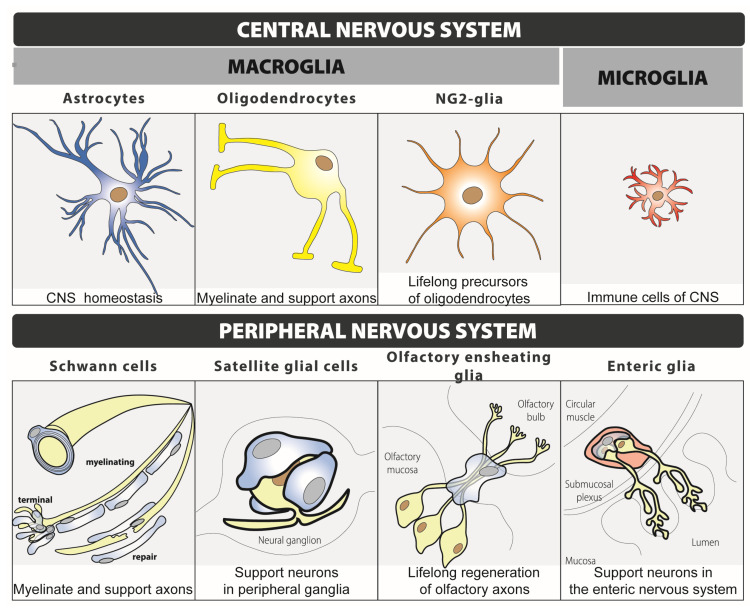
Classification and main function of glial cells.

**Figure 2 ijms-24-12553-f002:**
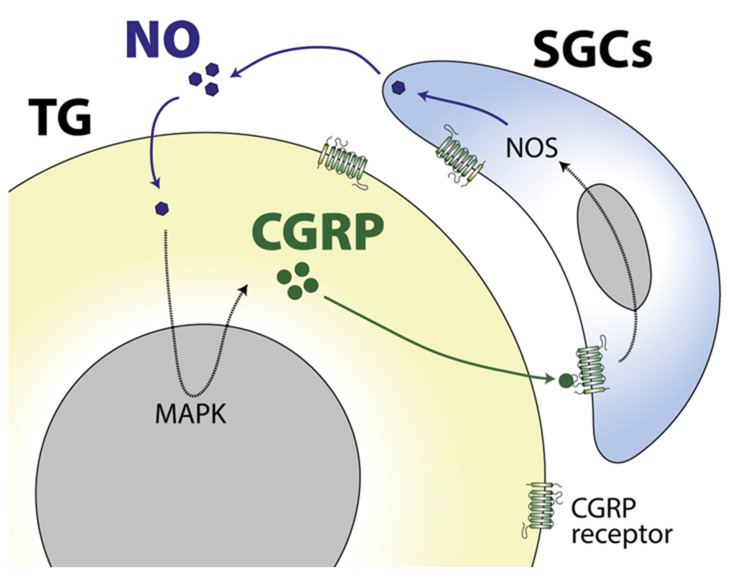
CGRP-mediated positive feedback between neurons and satellite glial cells of the trigeminal ganglia (TG). In certain conditions, neurons within the TG express and release CGRP to the extracellular space. Neuronal CGRP acts on the CGRP-receptor expressed in satellite glial cells (SGCs) inducing the expression and release of nitric oxide (NO) that, in turn, induces the expression of neuronal CGRP via MAPK signalling. This CGRP-mediated positive feedback may be involved in the chronification of migraine.

**Figure 3 ijms-24-12553-f003:**
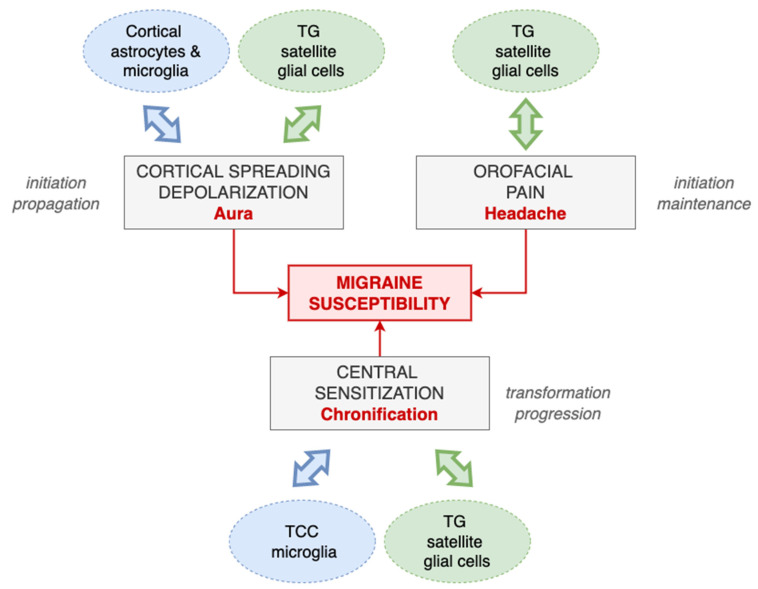
Summary of the potential role that glial cells may have in migraine. Data from preclinical studies suggest that glial cells may have an important impact on migraine susceptibility by modulating different phases of migraine, including aura, headache and migraine chronification. Cortical astrocytes and microglia and trigeminal ganglia (TG) satellite glial cells modulate the initiation and propagation and, at the same time, are modulated by cortical spreading depolarization. Satellite glial cells from the TG also modulate the initiation and maintenance of orofacial pain and central sensitization, where trigeminocervical (TCC) microglia also have a relevant role. Moreover, orofacial pain and central sensitization also impact the function of glial cells.

**Table 1 ijms-24-12553-t001:** Summary of the evidence that shows an implication of glia in different phases of migraine (CSD: cortical spreading depolarization; NTG: nitroglycerin; TG: trigeminal ganglia; TCC: trigeminocervical complex).

Migraine Phase	Glial Cell	Evidence of Implication	Proposed Function of Glia in Migraine	Ref.
Cortical spreading depolarisation (CSD)—aura	Astrocyte	Calcium waves inhibition blocks vascular changes without altering CSD propagation	Regulation of vascular response, but not the propagation, of CSD	[30,31,32]
	Astrocyte	Their regulation of extracellular glutamate concentration increases the susceptibility to CSD	Modulation of CSD initiation	[33,34,35]
	Astrocyte	Acute and chronic CSD induce astrocytosis, which is reverted after CSD termination	Implication in CSD	[36,37]
	Microglia	CSD activates, and induces the migration and motility of cortical microglia	Implication in CSD	[38,39,40]
	Microglia	In vitro depletion inhibits the induction of spreading depolarization	Essential for CSD initiation	[41]
	Satellite glial cell	Transcriptional activation in TG 1.5 h after induction of CSD	Implication in CSD	[42]
Orofacial pain—headache	Satellite glial cell	Expression of CGRP receptor in TG	Essential for headache induction	[43,44]
	Satellite glial cell	Expression of substance P receptor and PACAP and its receptors in TG	Regulation of headache induction via different neuropeptides	[44,45]
Chronification	Satellite glial cell	Existence of a CGRP-mediated positive feedback between them and neurons in TG	Potential implication in migraine chronification and target of anti-CGRP treatments	[46]
	Microglia	Activation in TCC in the NTG-induced chronic migraine mice	Modulation of central sensitization	[47,48,49,50]
	Microglia	Expression of inflammatory molecules in TCC in the NTG-induced chronic migraine mice	Modulation of central sensitization	[51]
	Microglia	Expression of microRNA miR-155-5p in the TCC of chronic migraine mice that is correlated with hyperalgesia levels	Modulation of central sensitization	[52]
	Microglia	Systemic minocycline reduces hind-paw allodynia and microglia activation in NTG-induced chronic migraine mice	Modulation of central sensitization	[47]
	Microglia	Systemic roxadustat reduces NTG-induced hyperalgesia, inflammatory cytokine levels and microglia activation in NTG-induced chronic migraine mice.	Modulation of central sensitization	[53]

**Table 2 ijms-24-12553-t002:** Summary of the current available glia-specific delivery systems and therapeutic approaches (ALS: amyotrophic lateral sclerosis; AAV vectors: adeno-associated viral vectors).

Approach	Type of Approach	Advantages	Limitations	Ref.
AAV vectors	Delivery system	Allows the delivery of gene therapy specifically into the glial cells of interest	Currently, they have only been used to deliver gene therapy in the nervous system to target neurons, not glia	[104,105]
Nanoparticles	Delivery system	Cross the blood–brain barrier after systemic administration and release the content into the glial cells of interest	They have not been used to deliver specific therapies in glial cells yet	[106]
Cell replacement	Therapeutic approach	Replacement of dysfunctional glia restores an appropriate homeostatic environment	It has been tested in neurons, not glia, for ALS treatment	[107,108]

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
