# Peer review of "The Role of Glial Cells in Different Phases of Migraine: Lessons from Preclinical Studies"

_ijms, 2023, doi:10.3390/ijms241612553_

Round 1

Reviewer 1 Report

This is an excellently written paper.  I have just a few comments:

1. The authors need to be careful in implying translatability from the preclinical to the clinical.  Indeed, the preclinical may be translatable, but there is no such evidence as of yet.  A few examples that may be over-reaching are on lines 342ff: "...TCC microglia have in the central sensitization of chronic migraine...".  It should be worded as "...TCC microglia potentially have in the central...".  Similarly, line 349 says "their states can modulate the susceptibility to cortical spreading depression, headache and migraine chronification;...".  The preclinical data support an affect in CSD, but we actually don't know if there is relevance to headache or chronification.  Another example is line 417, which should be "There is supporting evidence that glial cells may play a role in migraine pathophysiology..."

2. The authors mention that glial cells can secrete/release neurotransmitters (gliotransmitters), cytokines and growth factors (line 69, also echoed in line 158).  How is this accomplished?  Is this classical vesicular fusion with the cell membrane?  What do we know and what intracellular machinery is needed?  For instance, for Figure 2, how does the CGRP get out of the cell?  

3. I have an optional recommendation: as I was reading 1.2, I was thinking that you might make a table of the different characteristics of the types of glia that are outlined in the first 2 paragraphs of this section.  However, you have a figure.  You might still consider some type of table.

4. Line 407: the sentence is awkward.  The word "of" might need to be deleted.

The sentence starting on 409: the "...more specific therapeutic approaches such as viral vectors, nanoparticles and cell replacement therapy are needed...": in this sentence, the authors have mixed delivery systems (viral vectors, nanoparticles) and therapeutics (cell replacement therapy).  I think you want to separate out the therapeutics/therapeutic targets, and delivery systems.

One small redundancy: 2 sequential sentences (starting on line 418 and 422) have "have been reviewed here".  You can change one to "as reviewed".

Author Response

This is an excellently written paper.  I have just a few comments:

  1. The authors need to be careful in implying translatability from the preclinical to the clinical.  Indeed, the preclinical may be translatable, but there is no such evidence as of yet.  A few examples that may be over-reaching are on lines 342ff: "...TCC microglia have in the central sensitization of chronic migraine...".  It should be worded as "...TCC microglia potentially have in the central...".  Similarly, line 349 says "their states can modulate the susceptibility to cortical spreading depression, headache and migraine chronification;...".  The preclinical data support an affect in CSD, but we actually don't know if there is relevance to headache or chronification.  Another example is line 417, which should be "There is supporting evidence that glial cells mayplay a role in migraine pathophysiology..."

We would like to thank the reviewer for taking the time to review the manuscript and for providing valuable comments to improve it. We have replied to the comments following the list provided:

  1. We agree with this comment and have now changed the wording to avoid misleading the reader and not implying translatability from the preclinical to the clinical. We have done this in the following sentences:

- line 21: microglia in the trigeminocervical complex are involved in central sensitization, suggesting a role in chronic migraine.

- line 342: the relevance that TCC microglia potentially have in the central sensitization of chronic migraine

- line 349: their states could potentially modulate the susceptibility to cortical spreading depression, headache and migraine chronification

- line 417: There is supporting evidence that glial cells may play a role in migraine pathophysiology

  1. The authors mention that glial cells can secrete/release neurotransmitters (gliotransmitters), cytokines and growth factors (line 69, also echoed in line 158).  How is this accomplished?  Is this classical vesicular fusion with the cell membrane?  What do we know and what intracellular machinery is needed?  For instance, for Figure 2, how does the CGRP get out of the cell?  

Thank you for pointing this out. We have now added information regarding the release of molecules from glial cells:

Line 70: … glial cells are capable of releasing neurotransmitters and growth factors via classic fusion of secretory vesicles that…

  1. I have an optional recommendation: as I was reading 1.2, I was thinking that you might make a table of the different characteristics of the types of glia that are outlined in the first 2 paragraphs of this section.  However, you have a figure.  You might still consider some type of table.

Thank you for suggesting this. Rather than including the information in an additional table, we have included the characteristics of the types of glia on figure 1 to make it more visual for the readers.

Fig1_modified

  1. Line 407: the sentence is awkward.  The word "of" might need to be deleted.

We have now deleted the word “of” in this sentence:

Line 407: … in migraine patients would be crucial, as…

  1. The sentence starting on 409: the "...more specific therapeutic approaches such as viral vectors, nanoparticles and cell replacement therapy are needed...": in this sentence, the authors have mixed delivery systems (viral vectors, nanoparticles) and therapeutics (cell replacement therapy).  I think you want to separate out the therapeutics/therapeutic targets, and delivery systems.

We have now changed this sentence as suggested:

Line 411: …more specific delivery systems, such as viral vectors and nanoparticles, and therapeutic approaches, such as cell replacement therapy, are needed…

  1. One small redundancy: 2 sequential sentences (starting on line 418 and 422) have "have been reviewed here".  You can change one to "as reviewed".

We have now changed the second sentence to avoid the redundancy:

Line 424: As reviewed here, the majority of the preclinical studies performed to date have…

Reviewer 2 Report

In this manuscript, the authors provide a narrative overview of the role of glia in the different phases of migraine through analysis of studies.

Based on current evidence, glial cells have become a key player in the pathogenesis and chronification of migraine and may offer a future therapeutic option for migraine attacks.

The topic is timely and may attract much attention. However, in its current version, the manuscript has several limitations that should be addressed.

1. The migraine chapter (1.1 Migraine) should be expanded with more information about what we currently know about its pathomechanism.

Furthermore, a little more should be written about migraine, for example, what stages it has, what types exist, etc. A reader who is not familiar with the subject does not know this.

A few words about CSD should already be written here so that readers understand what it all has to do with migraines.

In addition, it can be briefly mentioned that glial cells may also play a role in migraine attacks.

2. 1.2. Glial cells chapter: It's a bit chaotic, and it should be summarized in an orderly manner, which glial cells are found where in the nervous system, and what their function is.

3. Chapter 3 (1.3. Overview of the role of glia in neurological diseases) is a bit off-topic for me. Based on the title and abstract, it does not fit into the manuscript, even if some diseases show comorbidity with migraine.

4. Chapter 5 (Targeting glia: potential future therapeutic and diagnostic opportunities for migraine) It would be worthwhile to summarize in a table the potential therapeutic solutions so far (involving glia), have phase studies have been done with them, what are their advantages, disadvantages, etc.

5. Animal and human data should be better separated.

6. A summary diagram of how glial cells relate to the pathomechanism of migraine would be useful. In what way can their functioning be influenced, thus controlling certain aspects of a migraine attack? 

7. References:  I recommend authors use more references to support their claims. In my opinion, less than 150 articles for a review paper are insufficient. Currently, authors cite only 109 papers. I believe that adding more citations will help to provide better and more accurate background to this study.

References are missing for the following paragraphs:

Lines 63-67 „Glial cells, also named glia or neuroglia, are the most abundant cells within the nervous system. Although initially they were considered as passive supporting cells for neurons, it is now well-accepted that they play active roles in the development and function 65 of the nervous system. Among other functions, they maintain neural homeostasis by nurturing and enhancing neuronal function and by keeping a proper chemical environment.” 

Lines 69-71 „Moreover, glial cells are capable of releasing neurotransmitters and growth factors that influence the activity of neurons or other cells, from the immune system for instance, and of communicating with each other via gap junctions and calcium waves.” 

Lines 78-85 „Although each type of glial cell participates in a myriad of functions, astrocytes are mainly involved in maintaining the homeostasis of the CNS. Oligodendrocytes and Schwann cells are in charge of myelinating and supporting axons, whereas NG2-glia are lifelong precursors of oligodendrocytes. Microglia are the immune cells of the CNS and satellite glial cells support neurons within peripheral ganglia. Finally, olfactory ensheathing glia are lifelong regenerators of olfactory axons and enteric glia are in charge of supporting neurons in the enteric nervous system of the gastrointestinal tract.” 

Lines 89-97 „Glia are responsible for maintaining an homeostatic environment within the nervous system and participate in several processes that are essential for its correct development and function. Moreover, glia suffer morphological, transcriptional and functional changes in disease, a process that has been extensively characterized in astrocytes (astrogliosis) and microglia (microgliosis), highlighting the existence of an important contribution of glia in neurological disorders. Actually, there is a growing body of evidence that shows the implication of the different types of glial cells in a wide range of pathological conditions, including neurodevelopmental, neurodegenerative and neuropsychiatric disorders and in different pain conditions.” 

Lines 320-331 „Roxadustat is a hypoxia-inducible factor-1α (HIF-1α) stabilizer that was found to reduce basal and acute NTG-induced hyperalgesia, to decrease inflammatory cytokine levels and to inhibit microglia activation, which was also measured through analysing Iba1 expression levels. However, an important limitation of using such drugs to modulate the activity of glial cells is that they are rather general glial inhibitors, so when systemically administered, it is not possible to discern which glial cells are mediating the studied mechanisms. For instance, the studies using minocycline or roxadustat focused only on analysing changes in TCC microglia, but did not explore other regions that are also relevant for migraine pathophysiology, such as the TG for example. Hence, they cannot rule out the possibility that the effects seen on allodynia might not be due to the effects of minocycline on the TCC, but on other regions of the central or peripheral nervous system. Furthermore, the mechanisms of action of such drugs remain some what uncertain”

Lines 381- 385 Actually, preclinical studies have successfully used nanoparticles to specifically deliver small interfering RNAs (siRNAs) or mRNAs to either down or up regulate the expression of a protein of interest in the targeted astrocytes. Although these studies seem promising for their potential as glial-specific drug delivery, further investigations are needed to refine and translate these tools as therapeutic options for neurological disorders including migraine.” 

Recommended references so that the authors can delve deeper into the topic and correct the manuscript accordingly:

https://www.ncbi.nlm.nih.gov/pmc/articles/PMC7962070/

https://journals.sagepub.com/doi/10.1177/0333102410375725?url_ver=Z39.88-2003&rfr_id=ori:rid:crossref.org&rfr_dat=cr_pub%20%200pubmed

https://pubmed.ncbi.nlm.nih.gov/19660121/

https://pubmed.ncbi.nlm.nih.gov/21719118/

https://pubmed.ncbi.nlm.nih.gov/27334137/

https://pubmed.ncbi.nlm.nih.gov/19036526/

https://www.frontiersin.org/articles/10.3389/fncel.2021.693095/full

https://onlinelibrary.wiley.com/doi/10.1002/glia.23874

https://pubmed.ncbi.nlm.nih.gov/35052756/

https://www.ncbi.nlm.nih.gov/pmc/articles/PMC8406410/

https://www.science.org/doi/10.1126/sciadv.aaz1584

https://link.springer.com/article/10.1007/s11064-022-03849-w

https://www.embopress.org/doi/full/10.15252/emmm.201505944

https://www.frontiersin.org/articles/10.3389/fnmol.2023.1219574/full

8, Abbreviations: The full form of abbreviations is required in the first place of appearance in the main body of the manuscript. After the first appearance, only the abbreviated form should be used. Please correct it.

Line 254, 255 substance P - the abbreviation was already introduced earlier (line 251 substance P (SP))

Line 291 trigeminocervical complex (TCC) - the abbreviation was already introduced earlier (line 43)

Line 330 central or peripheral nervous system -  he abbreviation was already introduced earlier (line 74 and 77)

Line 360 GFAP -  the abbreviation was already introduced earlier (Line 159), moreover, in line 403, the full form is written again, it is not necessary, the abbreviated form is sufficient.

Line 350 cortical spreading depression - cortical spreading depolarization is mentioned elsewhere, it should be unified 

Line 393 and 396 GDNF - the full form (Glial Cell Line-Derived Neurotrophic Factor) is not listed anywhere 

9. Language proofreading is recommended. 

Minor editing of English language required.

Author Response

We would like to thank the reviewer for taking the time to review the manuscript and for providing valuable comments to improve it.

We have replied to the comments following the list provided:

  1. The migraine chapter (1.1 Migraine) should be expanded with more information about what we currently know about its pathomechanism. Furthermore, a little more should be written about migraine, for example, what stages it has, what types exist, etc. A reader who is not familiar with the subject does not know this. A few words about CSD should already be written here so that readers understand what it all has to do with migraines. In addition, it can be briefly mentioned that glial cells may also play a role in migraine attacks.

We thank the reviewer for these comments. However, this review is not focused on migraine pathophysiology. Hence, we consider that the information included in section 1.1 is enough to provide a general understanding of the disease and to have a clear understanding of the other sections included in the review.

  1. 1.2. Glial cells chapter: It's a bit chaotic, and it should be summarized in an orderly manner, which glial cells are found where in the nervous system, and what their function is.

The information on the which glial cells, where and their function is now summarized on the new version of figure 1.

  1. Chapter 3 (1.3. Overview of the role of glia in neurological diseases) is a bit off-topic for me. Based on the title and abstract, it does not fit into the manuscript, even if some diseases show comorbidity with migraine.

Section 1.3 provides an overview of the role that glial cells have in neurological diseases in general, as a way to introduce the specific topic in migraine. We consider that it is relevant because the reader will understand that glial cells are key players in a wide variety of neurological diseases, not only in migraine.

  1. Chapter 5 (Targeting glia: potential future therapeutic and diagnostic opportunities for migraine) It would be worthwhile to summarize in a table the potential therapeutic solutions so far (involving glia), have phase studies have been done with them, what are their advantages, disadvantages, etc.

We have now included a table to summarize this information:

Table 2. Summary of the current available glia-specific delivery systems and therapeutic approaches (ALS: amyotrophic lateral sclerosis; AAV vectors: adeno-associated viral vectors).

Approach

Type of approach

Advantages

Limitations

Ref.

AAV vectors

Delivery system

Allows the delivery of gene therapy

specifically into the glial cells of interest

Currently, they have only been used to deliver gene therapy in the nervous system to target neurons, not glia

(104, 105)

Nanoparticles

Delivery system

Cross the blood brain barrier after systemic administration and release the content into the glial cells of interest

They have not been used to deliver specific therapies in glial cells yet

(106)

Cell replacement

Therapeutic approach

Replacement of dysfunctional glia restores an appropriate homeostatic environment

It has been tested in neurons, not glia, for ALS treatment

(107, 108)

  1. Animal and human data should be better separated.

As stated in the title and in all the sections of the manuscript, this review is based solely on preclinical data. We have not included human data in the review.

  1. A summary diagram of how glial cells relate to the pathomechanism of migraine would be useful. In what way can their functioning be influenced, thus controlling certain aspects of a migraine attack? 

We have now included a diagram in Figure 3 that summarizes the role that glial cells may have in migraine.

Figure 3

Figure 3. Summary of the potential role that glial cells may have in migraine. Data from preclinical studies suggests that glial cells may have an important impact on migraine susceptibility by modulating different phases of migraine, including aura, headache and migraine chronification. Cortical astrocytes and microglia and trigeminal ganglia (TG) satellite glial cells modulate the initiation and propagation and, at the same time, are modulated by cortical spreading depolarization. Satellite glial cells from the TG also modulate the initiation and maintenance of orofacial pain and central sensitization, where trigeminocervical (TCC) microglia also have a relevant role. Moreover, orofacial pain and central sensitization also impact the function of glial cells.

  1. References:  I recommend authors use more references to support their claims. In my opinion, less than 150 articles for a review paper are insufficient. Currently, authors cite only 109 papers. I believe that adding more citations will help to provide better and more accurate background to this study.

As the reviewer says, the threshold of using 150 references is a personal opinion and it is not a requirement for submitting reviews in this and, to our knowledge, in any of the journals. All our arguments have been appropriately backed up with relevant references. Hence, we consider that there is no need to include extra references that will not add more relevant information on the topic reviewed here.

The different paragraphs that are claimed to not include references, they do include them. However, we have chosen not to repeat the same reference in every single sentence, as the reader understands that the information comes from the previously cited reference.

8, Abbreviations: The full form of abbreviations is required in the first place of appearance in the main body of the manuscript. After the first appearance, only the abbreviated form should be used. Please correct it.

We would like to thank the reviewer for highlighting this and have now changed the abbreviations accordingly on:

Line 251: we have deleted the abbreviation of substance P and have only used the full form.

Line 291: Trigeminocervical complex has been changed for TCC.

Line 330: we have changed Central or peripheral nervous system for CNS or PNS.

Line 404: we have changed glial fibrillary acidic protein for GFAP

Line 352: cortical spreading depression has been changed for cortical spreading depolarization

Line 395: we have now included the full form of GDNF at first use of the acronym.

  1. Language proofreading is recommended.

Although the initial version of the manuscript was already proofread by a native English speaker, the new version has been proofread again.

Reviewer 3 Report

By providing updated information, this review focused on the possible contributions of glial cells to the pathogenesis of migraine. The authors also provided their viewpoints on glia cells as a potential therapeutic and diagnostic target of migraine. This review is suitable for both basic and clinical researchers who are interested in migraine study.

Author Response

We would like to thank the reviewer for taking the time to review the manuscript and for providing such positive feedback on it.

Round 2

Reviewer 2 Report

Although I can't entirely agree with everything, I accept the authors' answers.